# Salinity tolerance loci revealed in rice using high-throughput non-invasive phenotyping

Nadia Al-Tamimi[1], Chris Brien[2,3], Helena Oakey[1], Bettina Berger[3], Stephanie Saade[1], Yung Shwen Ho[1], Sandra M. Schmöckel[1], Mark Tester[1] & Sónia Negrão[1]

High-throughput phenotyping produces multiple measurements over time, which require new methods of analyses that are flexible in their quantification of plant growth and transpiration, yet are computationally economic. Here we develop such analyses and apply this to a rice population genotyped with a 700k SNP high-density array. Two rice diversity panels, *indica* and *aus*, containing a total of 553 genotypes, are phenotyped in waterlogged conditions. Using cubic smoothing splines to estimate plant growth and transpiration, we identify four time intervals that characterize the early responses of rice to salinity. Relative growth rate, transpiration rate and transpiration use efficiency (TUE) are analysed using a new association model that takes into account the interaction between treatment (control and salt) and genetic marker. This model allows the identification of previously undetected loci affecting TUE on chromosome 11, providing insights into the early responses of rice to salinity, in particular into the effects of salinity on plant growth and transpiration.

[1] King Abdullah University of Science and Technology (KAUST), Division of Biological and Environmental Sciences and Engineering (BESE), Thuwal 23955-6900, Saudi Arabia. [2] University of South Australia, Phenomics and Bioinformatics Research Centre, Adelaide, South Australia 5001, Australia. [3] University of Adelaide, Australian Plant Phenomics Facility, The Plant Accelerator, Urrbrae, South Australia 5064, Australia. Correspondence and requests for materials should be addressed to M.T. (email: mark.tester@kaust.edu.sa).

For more than half of the world's population, rice (*Oryza sativa* L.), the most salt-sensitive cereal[1–3], is a dietary staple. It is estimated that ∼20% of irrigated lands are affected by salt (http://www.fao.org/water/en/). For example, the Indo-Gangetic Basin in India and the Indus Basin in Pakistan suffer losses in rice yield as high as 45% and 36–69%, respectively, from soil salinity[1,4]. Moreover, climate change is foreseen to increase saltwater ingress in coastal regions of Southeast Asia, where rice is the primary cultivated crop[5]. With the global population rising, a 26% increase in rice yield is predicted to be required to meet global demands in the next 25 years[6]. Hence, there is a vital requirement to significantly increase rice productivity on salinized lands.

Exposure of plants to soil salinity rapidly reduces their growth and transpiration rates (TRs) due to the 'osmotic component' of salt stress (*sensu* Munns and Tester)[2], which is hypothesized to be related to sensing and signalling mechanisms[7]. Over time, toxic concentrations of $Na^+$ and $Cl^-$ accumulate in the cells of the shoot, known as the 'ionic component' of salt stress, which causes premature leaf senescence[2,8,9]. Both osmotic and ionic components of salinity stress are likely to impact yield. Despite significant advances in our understanding of the ionic components of salinity tolerance, little is known about the early responses of plants to salinity stress[7]. Therefore, the discovery of new quantitative trait loci (QTL) contributing to salinity tolerance, with a focus on the 'osmotic component', has the potential to substantially improve crop productivity.

The paucity of work on 'osmotic tolerance' is likely to be due, at least in part, to the need for the development of new methods for the analysis of plant growth and transpiration. Despite progress in analysing the image-based phenotyping data collected non-invasively with high time and spatial resolution, few statistical methods have accurately modelled plant growth and transpiration. Here we report a new statistical method for quantifying plant growth and transpiration using the data generated by high-throughput non-invasive phenotyping. We apply this method to precisely quantify the effects of salt stress on the growth and transpiration of rice plants.

A genome-wide association study (GWAS) was undertaken with the aim of identifying new loci that contribute to the early responses of rice to soil salinity. Two rice diversity panels, *indica* and *aus*, were phenotyped at The Plant Accelerator. Rice plants were grown in waterlogged conditions, to represent this aspect of irrigated rice fields that should be included in salinity tolerance studies, as the effects of hypoxia on salinity tolerance has been well documented[10]. In addition, we explored a model for analysing GWAS that enabled interactions between treatment groups (control and salt) and the genetic marker of interest. This new model substantially improves the detection of significant single-nucleotide polymorphisms (SNPs) that are specifically related to the treatment. By combining the analysis of high-throughput phenotyping data and GWAS, we investigated the effects of salinity on relative growth rate (RGR), TR and transpiration use efficiency (TUE), and identified several new salinity tolerance loci associated with these previously uncharacterized traits.

## Results

### *Indica* maintains growth better than *aus* in saline conditions.
To assess the early responses of two rice diversity panels to saline conditions, we exposed a total of 553 rice accessions (297 *indica* and 257 *aus* varieties) from 24-day-old plants to 150 mM NaCl. Over 13 days, several physiological responses of plants exposed to high salinity were monitored and compared with those of plants maintained in low salt concentrations. This was done using high-throughput, non-destructive imaging. From three RGB images (one top and two side views), we made daily measurements of the total number of pixels for each plant, as a proxy for shoot biomass. There are numerous mathematical models to describe growth curves[11–13], but most of these models make assumptions about the shape of the curve; for example, exponential growth models are typically used for young seedlings and short growth intervals. However, we observed that in our experiment, plant growth was neither exponential nor logistic throughout the imaging period, and in particular, salt-treated plants did not show exponential growth.

To avoid these erroneous assumptions and to accurately describe plant growth and to better estimate RGR, we fitted cubic smoothing splines (hereafter referred to as splines) to smooth the trend in the projected shoot area (PSA) for each plant. Examination of the plots of the smoothed values for PSA, absolute growth rate (AGR) and RGR (Supplementary Fig. 1) indicates that neither exponential nor logistic curves would accurately describe the growth of plants in this experiment, particularly in the case of the salt-treated plants. This approach has the advantage of making no *a priori* assumptions about the shape of the curve; to our knowledge, this is the first time that cubic smoothing splines have been used to provide an unbiased analysis of high-throughput phenotypic data. Although several decades ago, splines were fitted to data to characterize growth from destructive harvesting[14], they have seldom been used since. Shipley and Hunt[15] advocated their use for characterizing growth and Li and Sillanpää[12] suggested their application to describe complex growth trends[12,15].

We found that PSA strongly positively correlates with shoot biomass when using the squared Pearson correlation coefficient (for example, $r^2 = 0.945$ for *indica* and $r^2 = 0.91$ for *aus* in the northeast (NE) Smarthouse, using Pearson's correlation; Supplementary Fig. 2), confirming our experimental set-up as suitable to monitor plant growth. From the smoothed PSA, we were able to calculate RGRs and AGRs between imaging days (Supplementary Fig. 1). As expected, RGR decreases through time to a greater extent under saline conditions than in control conditions (Fig. 1a,b). More specifically, a rapid reduction in biomass production was observed immediately after salt application, suggesting that the rice plants responded to the 'osmotic component' of salt stress, before a build-up of salt in the leaves could impact plant growth, as occurs after several days, in the later 'ionic phase'[2].

Based on characteristic growth patterns of smoothed PSA, RGR and AGR, we separated the response of the rice plants to salinity stress into four intervals for further analysis (Fig. 1a,b; Supplementary Fig. 1a–f). In interval I, 2–9 days after treatment, AGR increases in control conditions and then plateaus. During interval II, 2–6 days after treatment, the RGR declines less rapidly under control and saline conditions than during interval III (6–9 days after treatment). During interval IV, 9–13 days after treatment, AGR increases less in control compared with salt-treated plants. The first day after salt treatment was excluded from further analyses because of reduced confidence in its values, inevitable from the spline fitting occurring across an interval that had clearly distinct properties.

We found that RGR was lower in the salt-treated accessions than in the control plants for both rice panels (Fig. 1). Comparing the percentage decrease in salt-treated plants relative to control plants at each interval, consistent with the results of Campbell *et al.*[16], we found that *indica* lines maintained better growth than *aus* lines (Fig. 1c,d; Supplementary Table 1); for example, in the interval 2–6 days after treatment, growth of *indica* due to salinity decreased by 21%, while that of *aus* decreased by 29%.

The early growth response index (EGRI) provides an estimate for the early responses of plants to salinity. For all earlier

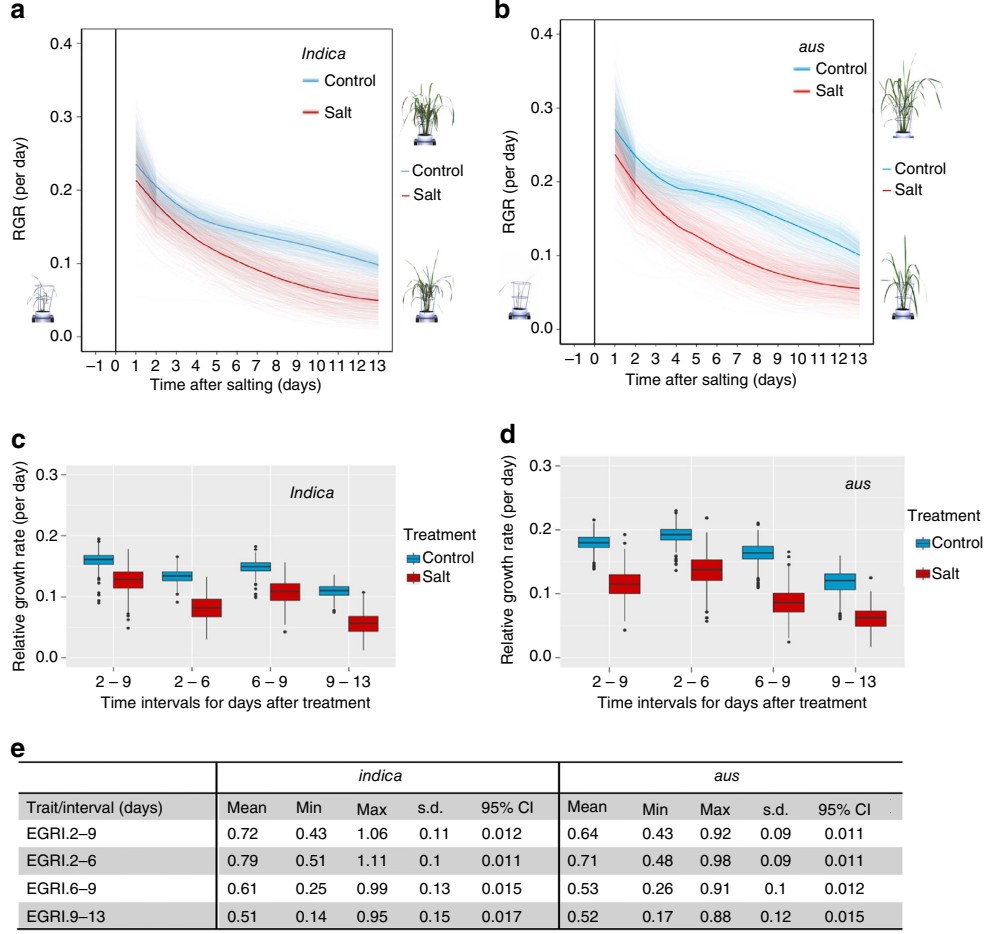

**Figure 1 | Relative growth rate (RGR) of salinity-induced responses comparing *indica* and *aus*.** (**a**) Smoothed RGR values were obtained from projected shoot area (PSA) values to which splines had been fitted, as shown in Supplementary Fig. 2. This was applied to the data from individual *indica* and (**b**) *aus* plants. The solid line represents the grand average of control conditions (blue) and saline conditions (red). In each panel, the RGB image of a rice plant on the left is representative of a plant 1 day before salt application. The RGB image on the top right side represents the same plant after 13 days of salt treatment, while the RGB image on the bottom right represents the same plant genotype at 13 days under control conditions. (**c**) Values of RGR at different time intervals for *indica* ($n = 528$; partially replicated; median = 0.13, 0.15, 0.11 and 0.10 for intervals: 2–9, 2–6, 6–9 and 9–13 days after salting, respectively) and (**d**) Values of RGR at different time intervals for *aus* ($n = 226$; fully replicated; median = 0.15, 0.17, 0.13 and 0.09 for intervals: 2–9, 2–6, 6–9 and 9–13 days after salting, respectively). (**e**) Table comparing the mean early growth response index (EGRI) at different time intervals for *indica* and *aus*. Min and max refer to the minimum and maximum means, respectively. s.d. refers to standard deviation. CI, confidence interval.

intervals, *indica* had a higher EGRI than *aus* (Fig. 1e). Although rice plants suffer a significant and rapid decrease in growth under salinity stress, *indica* accessions are better able to maintain their growth compared with *aus* accessions. In addition, accessions previously reported to be salt-tolerant, such as 'Pokkali' (an *indica* rice), have a higher EGRI than accessions previously reported as salt sensitive, such as 'IR28' (refs 17,18) (Supplementary Table 2). This result suggests that these early responses are likely to be an important component of overall plant salinity tolerance in the field.

**Indica maintains transpiration under saline conditions.** Our study also examines the previously unstudied traits of TR and TUE in response to moderate salinity stress under waterlogged growth conditions. To accurately describe TR, we fitted splines to daily measures of transpired water for each plant. As expected, we observed a clear acceleration in TR over time in control plants, and only a small increase over time in salt-treated plants (Fig. 2a,b). These results are consistent with a previous study of wheat and barley under saline conditions[19]. Notably, the *aus*

panel had a greater average decrease for TR in all four intervals when compared with *indica* (Supplementary Table 1). The substantial decrease in TR in the *aus* panel 9–13 days after treatment for both control and salt-treated plants can be explained by the sudden decrease in hours of sunshine at this point in the experiment. According to the Bureau of Meteorology's daily weather observations in March 2015, 9.1 h of sunshine on average were recorded during the first 10 days of imaging, but only 4.4 h of sunshine for the last 3 days (9–13 days after treatment).

TUE in this work is defined by the ratio of aboveground biomass produced per unit of water transpired and depends on the characteristics of the plants and on the environment where the plants grow. We calculated TUE as a third-order-derived trait (TUE is estimated from transpiration and growth from PSA, and these are, in turn, estimated from measures of water loss and pixel counts, respectively.) On average, TUE decreases marginally over time for control plants and more rapidly for salt-treated plants (Fig. 2c,d; Supplementary Fig. 3). Salinity reduced TR proportionally more than TUE, similar to wheat and barley[19]. The *indica* panel had a lower average decrease in TUE compared with the

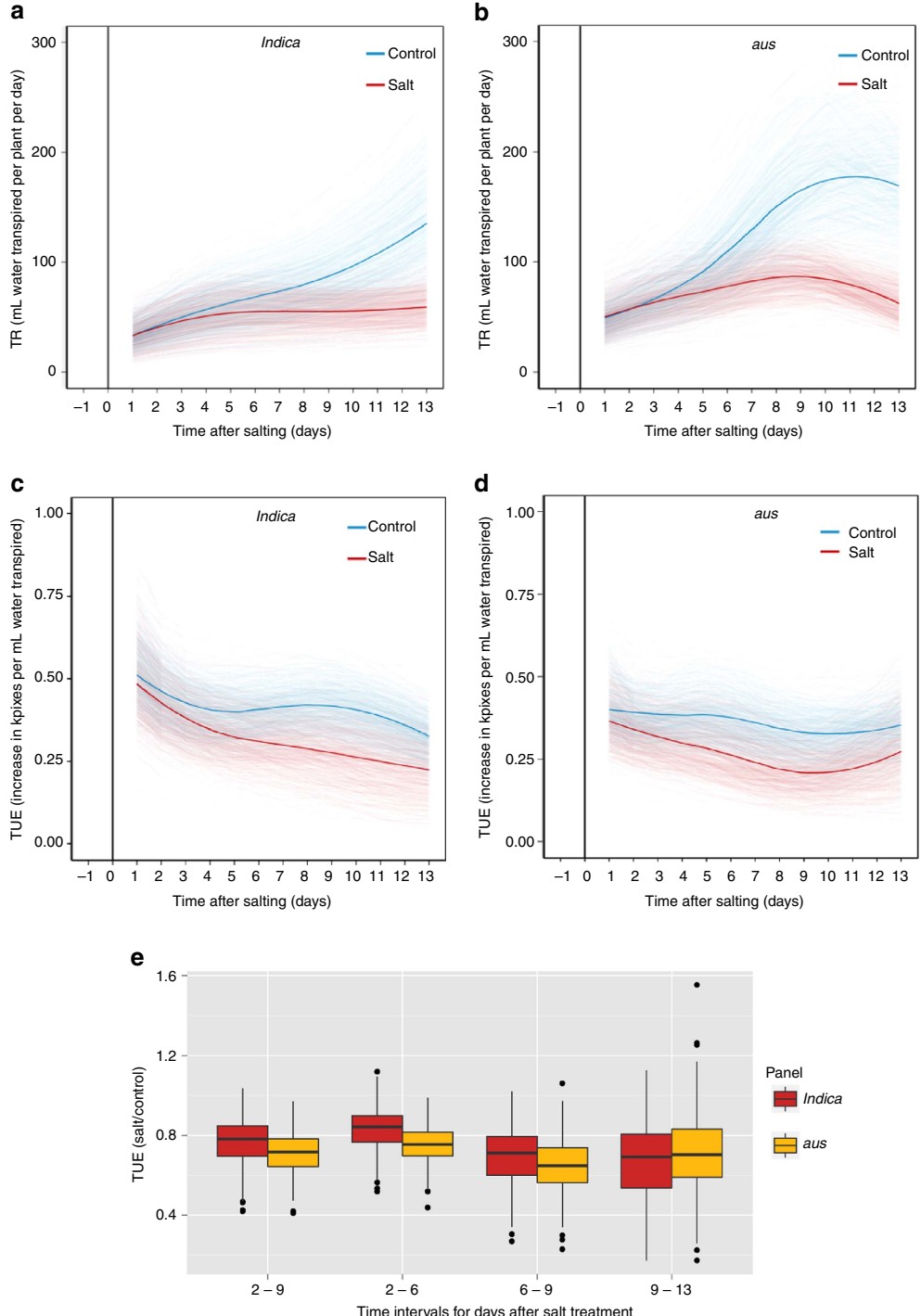

**Figure 2 | Transpiration of salinity-induced responses comparing _indica_ and _aus._** Spline curve fits of transpiration rate (TR) through time for individual (**a**) _indica_ and (**b**) _aus_ plants and transpiration use efficiency (TUE) through time for individual (**c**) _indica_ and (**d**) _aus_ plants. The solid blue lines represent the grand average spline in control conditions and the solid red lines represent the same in saline conditions. (**e**) Box plots of the TUE salinity tolerance index (salt/control), comparing _indica_ ($n = 528$; partially replicated; median = 0.78, 0.84, 0.71 and 0.69 for intervals: 2–9, 2–6, 6–9 and 9–13, respectively) and _aus_ ($n = 226$; fully replicated; median = 0.71, 0.75, 0.64 and 0.70 for intervals: 2–9, 2–6, 6–9 and 9–13 days after salting, respectively).

_aus_ panel (16.7% versus 24.4% for interval 2–6 days after treatment; Supplementary Table 1), and TUE was positively correlated with biomass production over time (RGR; Supplementary Fig. 4). To quantify the relative performance of plants with respect to TUE, we used a simple salt/control index[20] of the ratio of TUE in salt-treated plants relative to TUE in control plants over the same time period. A box plot of this index (Fig. 2e) shows that _indica_ tends to maintain a higher

salt/control index for TUE than _aus_ throughout the first three intervals.

**Association analysis of salinity-induced responses.** We used GWAS to identify genetic loci associated with the early responses of rice to salinity stress. We compared two sets of genotypic information for association analyses in the _indica_ panel—'GBS

44k SNP'[21] and a 'High-density rice array (HDRA) 700k SNP'[22]. Our results using the 'GBS 44k SNP' data set revealed no significant associations above the genome-wide significance threshold using the Bonferroni correction of $\alpha = 0.05$. In contrast, use of the recently generated data from the 'HDRA SNP' data set provided a high resolution of SNP detection, allowing the identification of multiple highly significant SNPs for both rice panels. Hence, using the high-resolution 'HDRA SNP' array provided a power gain for detecting genetic association loci.

In the association analyses, we used the traits RGR, TR and TUE at each interval. We initially performed GWAS using the conventional mixed linear model (MLM) with a kinship matrix between accessions to correct for genetic relatedness[23]. We then combined the responses in control and saline conditions using five derived indices for each main trait (RGR, TR and TUE) determined for each chosen time interval (2–9, 2–6, 6–9 and 9–13 days after treatment; Supplementary Table 3). This resulted in a total of 84 phenotypic traits for association analyses in each diversity panel and in the full population. To determine whether different loci were associated with trait variation in the different diversity panels, we used the *indica* and *aus* panels independently and also as a combined set named *INDAUS*. When using the conventional MLM, we found that no significant *P* values were obtained, whether using the Benjamini–Hochberg procedure, the false discovery rate or the Bonferroni thresholds. Even though previous studies have used a significance threshold of $P = 10^{-5}$ for the conventional MLM[24,25], we have retained this only as a suggestive threshold and present the MLM conventional results in Supplementary Fig. 5g–i and Supplementary Table 4. The conventional MLM results motivated us to explore a new association model, the interaction model, which integrates control and saline conditions. Using this model, the traits RGR, TR and TUE were investigated. The interaction model incorporates 'main effects' of the marker (SNP effect) and treatment (control or salt) as well as the marker-by-treatment interaction (SNP effect in response to the treatment—control or salt). The treatment effect for all responses and intervals was significant ($\alpha = 0.05$), reflecting the change in response due to the addition of salt. Significant loci found to be associated with one of the traits, in the marker term, suggests that the detected loci contribute to the traits regardless of the treatment, whereas significant loci in the marker-by-treatment term represent loci that are responsive to salinity treatment. The most significant phenotypic traits from our GWAS study are summarized in Supplementary Fig. 5 and Table 1.

Unique to this study is our assessment of a large number of genotypes for changes in TR and TUE immediately after salt imposition. Because TUE is a derived complex trait that has often been phenotyped using surrogate traits, genetic association with TUE has been challenging[13,26–28]. Nevertheless, we established a clear association between TUE and genetic loci. We observed an opposite dynamic of a QTL for TUE in *indica* (Fig. 3a), where the QTL on chromosomes 11 is strongest in the first interval after salt stress (2–6 days after treatment, $P = 3.03 \times 10^{-7}$), but not in the last interval (9–13 days after treatment).

Use of the interaction model led to the identification of several significant loci. When comparing the conventional model with our interaction model, we found a noticeably higher number of significant SNPs supporting each peak (increasing confidence in the association) and considerably smaller *P* values using our interaction model (Supplementary Fig. 5a–f; Table 1). In general, we found that the significant chromosomal regions in each diversity panel were different, indicative of large genomic variation between the two subpopulations, *indica* and *aus*. This result also suggests why there were no significant peaks in the association analyses in the combined *INDAUS*. In addition, there

was only a slight overlap of significant peaks for TUE in the marker-by-treatment term across three of the time intervals (2–6, 6–9 and 9–13 days after treatment; Fig. 3). The early time interval (2–6 days after treatment) had multiple significant peaks associated with TUE under saline conditions (on chromosome 11 of *indica* and chromosomes 5, 9 and 11 of *aus*), that disappeared from later time intervals (Fig. 3). We observed that in the marker term of *aus*, there were more common significant peaks during the two earlier intervals, while during the later interval, very different regions of the chromosome were significant. This suggests that the response of TUE is independent of salt treatment in the early intervals. The GWAS identified some loci present exclusively during the early time intervals after salt treatment, and the presence of other loci in only the later phase. Thus, rice exhibits a dynamic and complex response to salt stress, consistent with the hypothesis that there are two distinct phases of salt stress[2].

### Candidate genes underlying QTLs in early salinity responses.

We found association signals that are located close to specific genes of TUE for *indica* at SNP-11.3637597 on 36.3–36.4 Mb on chromosome 11 or in regions with high linkage disequilibrium (LD; such as TUE for *aus* at 23.6–24.2 Mb on chromosome 5). The LD region surrounding significant association peaks for TUE is presented in Table 1. The most promising candidate genes within the LD region were selected by excluding hypothetical genes and transposable elements. QTLs associated with TUE were examined in more detail because of the biological importance of this trait for crop improvement.

In the *indica* panel, one of the most promising regions is located at 3.62–3.76 Mb on chromosome 11. This region was detected by both the conventional MLM and our marker-by-treatment method. Although this region harbours several candidate genes, *Os11g07230* (encoding a receptor kinase) and *Os11g07240* (encoding a serine/threonine-protein kinase BRI1-like 2 precursor) are the most prominent candidates for this region. Another QTL is also found on chromosome 11 (at 2.78–2.79 Mb), between two candidate genes, *Os11g05930* (encoding a response regulator receiver domain-containing protein) and *Os11g05935* (encoding a mucin-type membrane protein involved in signal transduction). Note that *Os11g05930* is the orthologue to *HvPRR59* in barley, which has been associated with its early flowering time[29,30], while *Os11g05935* has an orthologue in *Caragana jubata*[31], which was found to respond to cold stress. These two salinity tolerance loci are the first evidence of this type of loci on chromosome 11 (ref. 32).

Using our marker-by-treatment term, we found that several regions significant for TUE traits in the *aus* panel under salt stress harbour genes related to signalling and signal transduction, such as *Os03g16130* (encoding a calcium/calmodulin-dependent kinase), *Os05g39870* (encoding OsCIPK28 and CAMK_KIN1, calcium/calmodulin-dependent protein kinase), *Os05g39900* (encoding a CBL-interacting serine/threonine-protein kinase 15), *Os05g46320* (encoding OsFBX173, an F-box domain-containing protein) and *Os05g47670* (containing a zinc-finger motif, a C3HC4-type domain-containing protein). Hence, our results show that candidate genes found in the early intervals, 2–6 and 2–9 days after treatment, are mainly encoding for signalling proteins, further supporting the hypothesis that the early responses to salt stress are related to signalling mechanisms.

### Discussion

To date, few genetic studies have used high time- and spatial-resolution non-destructive image-based phenotyping to address the complex response of plants to abiotic stresses[13,16]. In this

**Table 1 | Summary of candidate genes and local linkage disequilibrium region underlying the most significant using the interaction model**

| Chr. | Pop. | Time interval for days after treatment | QTL region (bp) | SNP ID | P-value | Candidate gene |
|---|---|---|---|---|---|---|
| 3 | *aus* | 2-6 | 8857549 | SNP-3.8856486. | $1.42 \times 10^{-8}$ | Os03g16070 (expressed protein) |
|  |  |  | 8878386..8907753 | in LD region |  | Os03g16334 (fringe-related protein) |
|  |  |  | 8891188..8907753 | in LD region |  | Os03g16120 (myosin heavy chain-related) |
|  |  |  |  |  |  | Os03g16130 (calcium/Calmodulin dependent kinase) |
|  |  |  |  |  |  | Os03g16140 (digalactosyldiacylglycerol synthase, chloroplast precursor) |
|  |  |  | 8907753..9120385 | in LD region |  | Region with several candidate genes |
|  |  |  | 8994756..9034370 | in LD region |  | Region with several candidate genes |
|  |  |  |  |  |  | Os03g16350 (DNA-binding protein) |
| 5 | *aus* | 2-6 | 3243210..3280158 | in LD region |  | Region with several candidate genes |
|  |  |  | 3280158..3307505 | in LD region |  | Region with several candidate genes Os05g06430 (OsPDIL2-1 protein disulfide isomerase PDIL2-1) |
|  |  |  | 8122862.. 8307465 | in LD region |  | Region with several candidate genes |
|  |  |  | 8917080..9056046 | in LD region |  | Region with several candidate genes |
|  |  |  |  |  |  | Os05g15890 (SNF2 family N-terminal domain containing protein) |
|  |  |  |  |  |  | Os05g15920 and Os05g15880(glycosyl hydrolase, putative expressed) |
| 5 | *aus* | 2-6 | 23407227..23536238 | in LD region |  | Region with several candidate genes |
|  |  |  |  |  |  | Os05g39850 (MCM3 - Putative minichromosome maintenance MCM complex subunit 3) |
|  |  |  |  |  |  | Os05g39870 (CAMK_KIN1 calcium/calmodulin dependent protein kinase) |
|  |  |  |  |  |  | Os05g39900 (CBL-interacting serine/threonine-protein kinase 15) |
|  |  | 2-9 and 6-9 | 23626677..24249943 | in LD region |  | Region with several candidate genes |
|  |  |  | 26826818..26934016 | in LD region |  | Region with several candidate genes |
|  |  |  |  |  |  | Os05g46320 (OsFBX173 - F-box domain containing protein) |
|  |  |  |  |  |  | Os05g46290 (T-complex protein, putative) |
|  |  |  |  |  |  | Os05g46350 (IQ calmodulin-binding motif domain containing protein) |
|  |  |  |  |  |  | Os05g46490 (hydrolase, alpha/beta fold family domain containing protein) |
|  |  |  | 27319587 | SNP-5.27256942. | $7.15 \times 10^{-7}$ | Os05g47670 (zinc finger, C3HC4 type domain containing protein) |
|  |  |  | 28178747 | SNP-5.28116101. | $4.16 \times 10^{-7}$ | Os05g49120 (NLI interacting factor-like phosphatase) |
| 8 | IND AUS | 2-6 | 10911095..11107666 | in LD region |  | Region with several candidate genes |
|  |  |  |  |  |  | Os08g18920 (protein kinase) |
|  |  |  |  |  |  | Os08g18740 (zinc knuckle-family protein) |
|  |  |  | 10745000..10844000 | in LD region |  | Region with several candidate genes |
|  |  |  | 13032730..13630347 | SNP-8.13627631. | $1.36 \times 10^{-6}$ |  |
|  |  |  | 20997000..21064000 | in LD region |  |  |
| 11 | Indica | 2-6 and 2-9 | 2789940..2795848 | SNP-11.2785843 | $3.54 \times 10^{-6}$ | Os11g05930 (response regulator receiver domain containing protein, expressed) |
|  |  |  |  | SNP-11.2791751 | $3.22 \times 10^{-7}$ | - |
|  |  | 2-6 | 3629008..3765215 | in LD region |  | Region with several candidate genes |
|  |  |  |  | SNP-11.3631311. | $3.03 \times 10^{-7}$ | Os11g07230 (receptor kinase) |
|  |  |  |  | SNP-11.3637597. | $3.03 \times 10^{-7}$ | Os11g07240 (serine/threonine-protein kinase BRI1-like 2 precursor) |
|  |  |  |  | in LD region |  | Os11g05930 (response regulator receiver domain) |
|  |  |  |  |  |  | Os11g05935 (mucin) |

Chr., chromosome; GWAS, genome-wide association study; LD, linkage disequilibrium; MLM, mixed linear model; Pop,, population QTL, quantitative trait loci; SNP, single-nucleotide polymorphism. All the significant associations are presented for the three diversity panels (*indica, aus* and *INDAUS*) and intervals (2-9; 2-6; 6-9; 9-13 days after treatment). Statistical significance of the GWAS associations were determined using the Bonferroni-adjusted threshold of $\alpha = 0.05$, which corresponded to $P = 8.99 \times 10^{-6}$, $2.57 \times 10^{-6}$ and $3.02 \times 10^{-6}$ for the *INDAUS, indica* and *aus* sub-populations respectively.

work, we have demonstrated the effectiveness of high-throughput phenotyping for dissecting the genetic architecture of complex traits such as RGR, TR and TUE, and the effects of salinity on these parameters. Previous studies have used high-throughput phenotyping to elucidate dynamic responses of rice associated with growth and morphology and in response to salinity[16]. To analyse plant growth using high-throughput phenotyping, we have tested several statistical approaches.

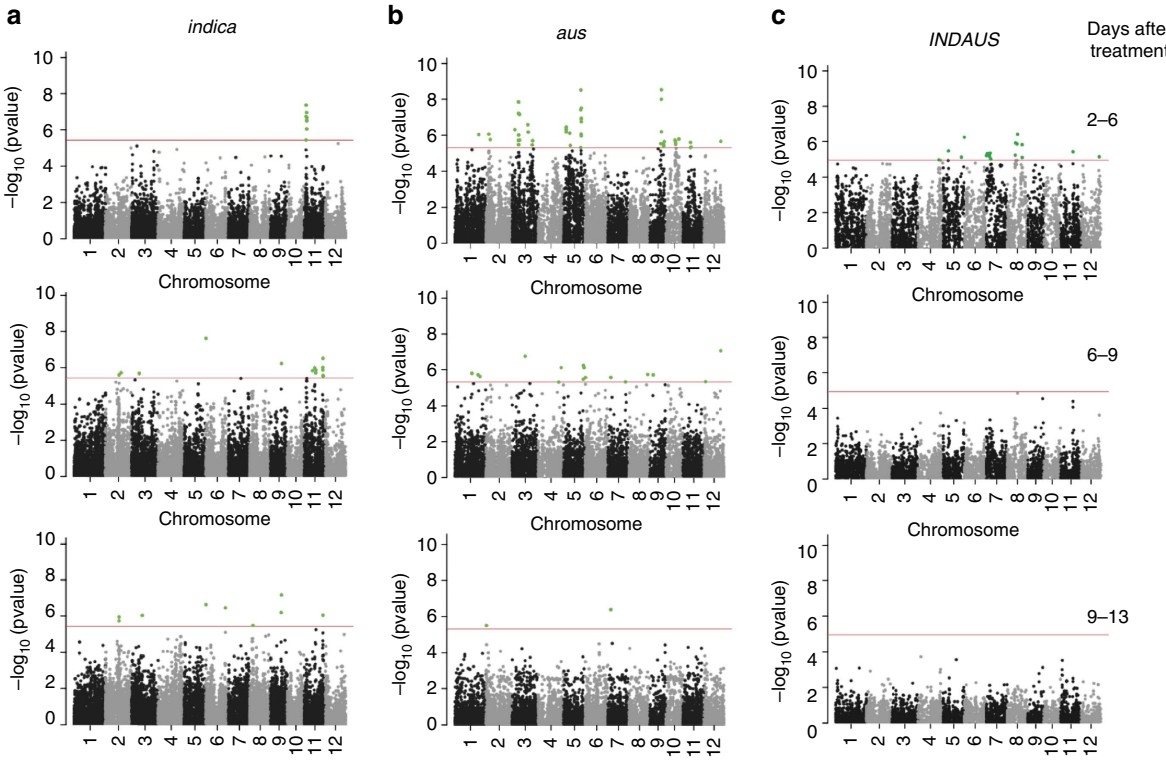

**Figure 3 | Marker-by-treatment interaction model using transpiration use efficiency in response to salinity.** SNPs are highlighted in green if they reach genome-wide significance for association with TUE at each time interval in (**a**) *indica*, (**b**) *aus* and (**c**) *INDAUS*. SNPs associated with TUE are shown at the different time intervals: 2–6, 6–9 and 9–13 days after treatment (panels top to bottom). Horizontal red lines indicate Bonferroni-adjusted threshold of $\alpha = 0.05$, which corresponded to $P = 8.99 \times 10^{-6}$, $2.57 \times 10^{-6}$ and $3.02 \times 10^{-6}$ for the *INDAUS*, *indica* and *aus* subpopulations, respectively.

First, a method for calculating the growth and transpiration must be chosen. For young seedlings and short growth intervals, exponential growth models have been used[10], while others have used a logistic model[12,13,33]. In particular, the use of a longitudinal model based on a decreasing logistic model to estimate rice growth under saline conditions has clearly shown the high statistical power of high-throughput phenotyping[16]. The fitting of such curves has the advantages of allowing extrapolation and characterization of growth using the few parameters associated with the particular curve. However, growth does not always conform to a particular pre-determined equation[15]. For example, the exponential growth model assumes a constant RGR, but this assumption was clearly not met in our experiment (Supplementary Fig. 1); similarly, logistic curves were not appropriate to describe our data. Splines have been suggested when more complex or unbiased models are required[12]. Splines have the advantage of making no *a priori* assumptions about the shape of the curve and allow us to faithfully follow the growth and transpiration trends in an unbiased way, while removing transient influences on the growth (such as a shady day). Furthermore, as we describe next, it is possible to characterize growth in terms of a few parameters calculated from splines. To our knowledge, this is the first time that splines have been used to analyse high-throughput phenotypic data.

Second, using a longitudinal model in a joint analysis of all plants consumes high-computational resources, encouraging the development of more efficient methods for analysing high-throughput phenotyping data. In this study, we used splines to establish the trends in growth independently for each individual plant; this was a purely descriptive curve-fitting procedure that requires minimal computing resources. Growth rates over time were then calculated from the fitted splines and time intervals

were identified and selected based on knowledge of the underlying biology and/or from characteristic growth patterns. Measures of plant growth and transpiration were obtained for these intervals and each formed a single trait for which there was one value for each plant. This calculation of measurements for a set of time intervals is a unique aspect of the approach that we describe here, although our method harks back to the approach of Rowell and Waters[34]. Each of these traits was then adjusted for spatial variation using a mixed model and subjected to GWAS analysis. By reducing the traits to one value per plant, the need for excessive computational resources was prevented. Furthermore, by combining this method of calculating traits with our interaction model for association analysis, we were able to detect previously undescribed loci associated with salinity response in rice.

We also present a new association model that takes into account the interaction between treatment (control and salt) and the genetic marker. The use of this interaction model enables identification of significant loci specifically associated with salinity stress. The interval 2–6 days after treatment represents the early response phase to salinity, while the later intervals, 6–9 and 9–13 days after treatment, appear to be affected by different processes, which may represent a shift in the relative importance of 'osmotic' and 'ionic' processes between these intervals. This is suggested by the presence of QTLs that have a stronger effect during the first interval, 2–6 days after treatment, which disappear after 6 days after treatment; for example, the QTL associated with TUE on chromosome 5 in the *aus* panel, specific to salinity response in our marker-by-treatment interaction model, appears likely to correspond with the early responses to salinity. Consistent with these findings, 2–6 days after treatment, we found that a major involvement of signalling genes in

response to salt, as previously hypothesized by others[2,7,35]. It should be said that, although a QTL is not resolved after 6 days, this does not necessarily mean the process has disappeared—but simply that the process is less easily measured because other processes have increased in significance. That the early phase is likely to be independent of shoot ion accumulation is supported by the absence of QTLs identified in previous studies[36] that affect leaf ion accumulation, in particular the locus containing *OsHKT1;5*, on chromosome 1.

Rice sensitivity to salinity varies according to growth stage, being particularly sensitive at both seedling and reproductive periods[32]. Nevertheless, yield is also known to be affected by exposure of the vegetative stage to salinity, affecting parameters such as tiller number per plant and duration of stress[37–39]. Previous studies in rice have suggested the use of several selection criteria to increase salinity tolerance[37,40]. In this study, we quantified the effects of salinity on RGR, TR and TUE in plants growing at their vegetative stage. Advances in image-based phenotyping that use new methods to recover the structure of a plant and build a three-dimensional (3D) model[41] are likely to play a useful role in determining tiller number per plant, a key trait for rice yield. Our results suggest that maintenance of TUE under salinity is an important process contributing to salinity tolerance during the main vegetative stage of plant growth. The new salinity tolerance loci presented in this study could be useful in breeding programs to improve rice productivity on salinized lands.

## Methods

**Plant material growth conditions and salt treatment.** In our study, we used two diversity panels composed of two major rice subpopulations, *indica* and *aus*, which contain 297 and 257 accessions, respectively. These panels were selected by breeders at the International Rice Research Institute and collaborating partners in the context of the 'Phenomics of Rice Adaptation and Yield Potential' (PRAY) project, funded by the Global Rice Science Partnership (http://ricephenonetwork.irri.org/). The two Phenomics of Rice Adaptation and Yield Potential panels represent sub-species level genetic diversity, including landraces of different geographic origin and agro-ecological adaptation and mega-varieties that are cultivated in vast areas. Information about the accessions, including genotype name and Genebank accession code, can be found in Supplementary Table 5.

We phenotyped both rice diversity panels at The Plant Accelerator (Australian Plant Phenomics Facility, University of Adelaide, Adelaide, Australia; − 34.97113, 138.63989) between January and March 2015. This experiment used two quarantine-approved greenhouses fitted with conveyor systems, described as the NE and northwest (NW) Smarthouses. Throughout the two experiments, the temperature in the greenhouses was set to 28 °C during the day and 26 °C at night. Relative humidity was increased using three humidifiers per room (Rotation Atomizer Defensor ABS3, Condair, Switzerland) and reached an average of 66% over the growing period in the NE Smarthouse and 62% in the NW Smarthouse. Rice seeds were treated according to Australian quarantine guidelines by immersion in water at 57 °C for 15 min followed by surface sterilization with Thiram fungicide and germinated on moist paper towels in plastic boxes for 4 days. Three uniformly germinated seeds of each genotype were transplanted into soil. A blue pot (125 mm diameter, 137 mm height) with drainage holes containing 1.35 kg of U.C. mix[42] and fertilizer (1.5 kg Mini Osmocote per 600 liter U.C. soil base) was placed inside a white pot that had a closed base (140 mm diameter, 193 mm height) sitting on top of a sealed plastic container (93 mm diameter, 50 mm height) as detailed in Supplementary Fig. 6. This system simulated the waterlogged conditions of rice fields, but prevented water from spilling onto the conveyor system. Plants were thinned to one uniformly sized seedling per pot 11 days after transplanting (DAT). The soil surface was covered with 200 g of white gravel (particle size ∼2–5 mm) 14 DAT to reduce algal growth and minimize water evaporation from the soil. For the first 17 DAT, the pots were watered daily to an approximate 900 ml water level to ensure that all plants had exactly the same pot + soil + water weight, which further allowed the estimation of water loss for each plant during the experiment. At 17 DAT, the pots were manually loaded onto the conveyer system where blue poly-vinyl chloride mats were placed on top of the gravel to further reduce soil evaporation while providing a favourable background colour for image analysis. A blue carnation frame was placed in the pot to support the plant. Water levels were monitored and adjusted daily by the Scanalyzer 3D system weighing and watering system (LemnaTec GmbH, Aachen, Germany). Watering levels were kept the same for control and salt-treated plants due to the

small relative differences in the overall shoot biomass between the treatments, especially when compared with the overall pot weight.

Salt treatment was applied at 24 DAT (29 days after germination) by adding 100 ml of 1.425 M NaCl to the bottom of the outer white pot to a final concentration of 150 mM NaCl in the soil solution after drying down to 950 ml. Control plants received 100 ml of water on the same day. The concentration of salt in the soil was maintained at constant levels by watering each pot to a target volume of 950 ml. Daily imaging and watering were continued for 13 days after treatment until 37 DAT. A time-lapse video of a randomly selected genotype shows plant growth in control and saline conditions (Supplementary Movie 1).

**Experimental design.** Experiments were performed concurrently in NW and NE Smarthouses (Supplementary Fig. 7). A split-plot design was used to assign the lines and conditions to 1,056 plant carts (that is, pots). Each Smarthouse contained 528 plant carts, with each cart holding one pot with a single plant, arranged in 24 lanes by 22 plant carts (Supplementary Fig. 8). Pairs of consecutive carts in a lane, referred to as a main plot, contained a control (no salt) and a salt-treated plant with the same genotype (assignment of control and salt treatment was randomized for each pair). The design of the main-plot for the *indica* panel was a partially replicated, blocked, row-and-column design; the design of the main plot for the *aus* panel was an unequally replicated, blocked, row-and-column design. In both cases, each block contained the zones of four consecutive lanes. The design of the main plot was generated using *DiGGer*[43], a package for the R statistical computing environment[44]. In addition, there were 24 evaporation carts (that is, pots without a plant), one per lane interspersed between main plots and treated equally to the 528 occupied plant carts (Supplementary Fig. 9a). These evaporation carts were used to determine the spatial variation of evaporation within the Smarthouses and to calculate the transpiration of plants (which equates to the water loss through evaporation subtracted from the total water loss per plant cart). Their locations were chosen to minimize their overall distance from plant carts.

**RGB image capture and image analysis.** Plant imaging started at 23 DAT and continued until 37 DAT (which was 13 days after treatment). Shoot images were taken using the LemnaTec 3D Scanalyzer system (LemnaTec GmbH, Aachen, Germany) at The Plant Accelerator. Plants were imaged daily in an imaging chamber using two 5-megapixel visible/RGB cameras (Basler Pilot piA2400-17gm). Three images were taken per plant, two images from the side at a 90° rotation and one from the top. All captured images were analysed using the LemnaTec Grid software package (LemnaTec GmbH, Aachen, Germany). In brief, digital image processing consisted of a nearest-neighbour colour classification for foreground-background separation, noise reduction and object composition (Supplementary Fig. 10). Non-plant pixels, such as pots, support frames or poly-vinyl chloride mats (background), were removed to extract only plant pixels (object). The PSA was extracted from all three RGB images, and the sum of PSA from all three images was used to estimate shoot biomass as previously described[16,45]. The complete RGB data set for *indica* and *aus* panels is available online through The Plant Accelerator data portal.

**Data preparation for main traits and their derived indices.** Data corresponding to the PSA, the weight before imaging, the weight after imaging and the weight of the evaporation cart were recorded daily between 23 and 37 DAT. One exception was caused by an equipment malfunction on DAT 35 for the *aus* panel, which prevented weighing on this day. All subsequent analyses were performed on plants between 26 and 37 DAT, corresponding to 2–13 days after treatment. Note that we excluded day 1 after salt treatment (DAT 25) from the analyses due to a marked fall in the RGR from 1 to 2 days after treatment, caused by the reduced confidence in their values after spline fitting.

Using the R package *imageData*[46] 0.1–13 (http://cran.r-project.org/web/packages/imageData/index.html), the data were input into the R statistical computing environment[44] and processed to produce the traits for statistical analysis. The PSA was estimated using the sum of all plant pixels from all three RGB images (two lateral and one top) as previously described[16,45]. The TR was calculated from the transpiration and evaporation, for each plant and for each day according to the following equation:

$$\mathrm{TR}_{(t_{k-1}, t_k)} = \frac{T_{(t_{k-1}, t_k)}}{t_k - t_{k-1}} = \frac{(\mathrm{WA}_{t_{k-1}} - \mathrm{WB}_{t_k}) - E_{(t_{k-1}, t_k)}}{t_k - t_{k-1}},$$

where $T$ is transpiration, WA is weight after imaging, WB is weight before imaging and $E$ is the weight of evaporation cart. In this equation, $t$ is time, $k$ refers to the beginning of a time interval of interest and $k-1$ refers to the previous day. The weight of the evaporation cart was calculated in the same way as transpiration, but from the weight after imaging and weight before imaging for the nearest evaporation cart (Supplementary Fig. 9b). Estimates of PSA and transpiration were used to calculate AGRs, RGRs and TUE between two time points, $t_k$ and $t_j$, as

follows:

$$\text{AGR}_{(t_j,t_k)} = \frac{\text{PSA}_{t_k} - \text{PSA}_{t_j}}{t_k - t_j}, \quad \text{RGR}_{(t_j,t_k)} = \frac{\ln(\text{PSA}_{t_k}) - \ln(\text{PSA}_{t_j})}{t_k - t_j} \quad \text{and}$$

$$\text{TUE}_{(t_j,t_k)} = \frac{\text{PSA}_{t_k} - \text{PSA}_{t_j}}{\sum_{h=j+1}^{k} T_h}.$$

The calculation of the above-described parameters (PSA, AGR and RGR) allowed the identification of plants with anomalous growth patterns. The following criteria were used to exclude plants from subsequent analyses: (i) dead or dying plants, (ii) plants that were not loaded; (iii) plants that missed the correct salt treatment; and (iv) salt-treated plants that markedly outperformed their associated control plant. After removing these plants, PSA and TR values were smoothed by fitting a cubic smoothing spline to the data for each remaining plant[15]. Smoothed AGRs, RGRs and TUEs between imaging days were calculated from the smoothed PSA and TR. Based on plots of the smoothed PSA and their corresponding AGRs and RGRs (Supplementary Fig. 1), we decided to calculate AGRs, RGRs and TUEs for the following intervals: 2–9, 2–6, 6–9 and 9–13 days after treatment. These intervals were considered to be appropriate given the growth patterns observed in the plots. Also, there appears to be changes in the behaviour of the plants between 6 and 9 days after treatment.

Five derived indices were calculated for each of the three main traits (RGR, TR and TUE) at each chosen interval, as described in Supplementary Table 3.

The EGRI was calculated for each interval as the RGR in salt divided by the RGR in control for the pair of plants forming a main plot.

**Spatial correction of phenotypic analysis.** To produce phenotypic means adjusted for the spatial variation in the Smarthouses, a MLM analysis was performed for the AGRs, RGRs, TRs and TUEs from each interval. The maximal MLM for this analysis was calculated from the formula:

$$\mathbf{y} = \mathbf{X}\boldsymbol{\beta} + \mathbf{Z}\mathbf{u} + \mathbf{e},$$

where $\mathbf{y}$ is the response vector of values for the trait being analysed, $\boldsymbol{\beta}$ is the vector of fixed effects, $\mathbf{u}$ is the vector of random effects and $\mathbf{e}$ is the vector of residual effects. $\mathbf{X}$ and $\mathbf{Z}$ are the design matrices corresponding to $\boldsymbol{\beta}$ and $\mathbf{u}$, respectively.

For the maximal MLM, the fixed effect vector, $\boldsymbol{\beta}'$, is partitioned as follows:

$$[\mu \; \boldsymbol{\beta}'_S \; \boldsymbol{\beta}'_{S:xZ} \; \boldsymbol{\beta}'_{S:xP} \; \boldsymbol{\beta}'_L \; \boldsymbol{\beta}'_T \; \boldsymbol{\beta}'_{L:T}]$$

where $\mu$ is the overall mean and $\boldsymbol{\beta}_S$ are the vectors of: Smarthouse effects, linear coefficients for trend within each Smarthouse over the zones, linear coefficients for trend within each Smarthouse over the east-west positions of main plots, line fixed effects, treatment fixed effects and fixed effects for line-treatment combinations, respectively.

Also, the random effects vector, $\mathbf{u}'$, is partitioned as follows:

$$[\mathbf{u}'_{S:sZ} \; \mathbf{u}'_{S:sP} \; \mathbf{u}'_{S:dZ} \; \mathbf{u}'_{S:dP} \; \mathbf{u}'_{S:Z:M}]$$

where the $\mathbf{u}_S$ are the vectors of the following: coefficients of the spline basis functions for fitting smooth trends within each Smarthouse over zones, coefficients of the spline basis functions for fitting smooth trends within each Smarthouse over the east-west positions of main plots, deviations within each Smarthouse from the smooth trend over zones, deviations within each Smarthouse from the smooth trend over the main-plot positions and main-plot random effects within each zone within each Smarthouse, respectively. The design matrices $\mathbf{X}$ and $\mathbf{Z}$ are partitioned to conform to the partitioning of $\boldsymbol{\beta}$ and $\mathbf{u}$, respectively. It is assumed that each subvector of random effects, $\mathbf{u}_i$, is distributed $N(\mathbf{0}_{m_i}, \sigma_i^2 \mathbf{I}_{m_i})$, where $\mathbf{0}_{m_i}$ is the $m_i$-vector of zeroes, $\sigma_i^2$ is the variance of the $i$th set of random effects, $\mathbf{I}_{m_i}$ is the identity matrix of order $m_i$, and $m_i$ is the order of $\mathbf{u}_i$. Further, with $\mathbf{y}$ being ordered so that all observations for the control treatment are followed by those for the salt treatment, the distribution of the residual effects $\mathbf{e}$ is assumed to be:

$$N\left(\mathbf{0}_{1056}, \begin{bmatrix} \sigma_C^2 & 0 \\ 0 & \sigma_S^2 \end{bmatrix} \otimes \mathbf{I}_{528}\right)$$

where $\sigma_C^2$ are $\sigma_S^2$, respectively, the variances of the residuals for the control and salt treatments, meaning that this model for the residuals allows for the two treatments to have different residual variances. Except for $\sigma_{S:Z:M}^2$, each of the variance components and the need for unequal residual variances were tested via restricted maximum likelihood estimation (REML) ratio tests with using ASReml-R[47] and ASRemlPlus[48], packages for the R statistical computing environment[44]. The nonsignificant terms were removed from the model and the phenotypic means were obtained using the resulting model.

The MLM analysis for EGRI was based on an MLM derived from the model above, by removing the terms involving Treatments. In the case of RGRs, TRs and TUEs, REML ratio tests were conducted to test for unequal condition variances. For all traits, REML ratio tests were used to test for zone and main-plot deviations and curved trends. The fitted model reflected the results of these tests. Wald $F$-tests were conducted for linear trends and terms involving the terms lines and conditions from the equations. From these analyses, the best linear unbiased estimates were obtained. RGRs, TRs and TUEs were obtained for the line-treatment (that is, genotype-treatment) combinations and the EGRI indices were obtained for each genotype.

At each time interval, correlation analysis was performed for the main traits (RGR, TR and TUE) with the Pearson option using the function *chart.Correlation()* from the package *PerformanceAnalytics* a package for the R statistical computing environment[44,49] (Supplementary Fig. 4).

**Genome-wide association analysis.** Association analyses were performed with the two diversity panels (*indica* and *aus*) independently and also with a combined single set (*INDAUS*). Genotypic data were composed of the HDRA data set, a 700k SNP array designed by McCouch's laboratory at Cornell University[22]. From the HDRA SNPs, only SNP markers with a minor allele frequency of ≥0.05 and a number of accessions with a minor allele >6 were used for association analyses. This resulted in a total of 397,659 SNPs in the *indica* subpopulation, 394,785 SNPs in the *aus* subpopulation and 304,877 SNPs in the combined *INDAUS* subpopulations for GWAS. This project also compared two sets of genotypic information in the *indica* panel, -'GBS 44k SNP'[21] and 'HDRA SNP'[13,22] (Supplementary Table 6).

We used two GWAS approaches: one considers the response for each salt treatment in a separate model, and the other, referred to also as the interaction model, amalgamates each response for both treatments. Both approaches use an MLM that is fitted to include the kinship (**K**) matrix as a random effect to control for population structure[50,51]. When analysing the combined population *INDAUS*, we also included principal components to control for population structure, because of the deep population structure in rice[21,23]. Thus, the difference between the two approaches is the ability to examine the interaction effects between the SNP markers and the salt treatment, which is included as a cofactor in the interaction model. Analyses of the first approach were conducted using TASSEL software (http://www.maizegenetics.net/tassel)[52]; analyses of the second approach, the new MLM referred to as interaction model, was fitted using the *ASReml-R*[47], a package for the R computing environment[44], according to the following equation:

$$\mathbf{y} = \mathbf{X}\boldsymbol{\beta} + \mathbf{S}\boldsymbol{\alpha} + \mathbf{T}\boldsymbol{\gamma} + \mathbf{Z}\mathbf{u} + \mathbf{e}$$

Here $\mathbf{y}$ is the response vector of phenotypic means for each line from both control and saline conditions, $\boldsymbol{\beta}$ is the vector of fixed treatment terms that fits an overall mean for both the control and saline conditions, and in the *INDAUS* combined population principal component cofactors, $\boldsymbol{\alpha}$ is the vector of fixed marker effects ($\boldsymbol{\alpha}$ and $\boldsymbol{\beta}$ terms are treated as factors with corner point constraints), $\boldsymbol{\gamma}$ is the vector of fixed interaction effects between the treatment and marker, $\mathbf{u}$ is the random vector of line effects and $\mathbf{e}$ is the vector of residual effects. $\mathbf{X}$, $\mathbf{S}$, $\mathbf{T}$ and $\mathbf{Z}$ are the design matrices corresponding to $\boldsymbol{\beta}$, $\boldsymbol{\alpha}$, $\boldsymbol{\gamma}$ and $\mathbf{u}$, respectively. The variance of the line random effects are $\text{Var}(\mathbf{u}) = \mathbf{K}\sigma_g^2$, where $\mathbf{K}$ is the genomic relationship matrix calculated using TASSEL software and $\sigma_g^2$ is the genetic variance; $\text{Var}(\mathbf{e}) = \mathbf{I}\sigma_e^2$, where $\mathbf{I}$ is the identity matrix and $\sigma_e^2$ is the residual variance. This new model was used for the three main traits, RGR, TR and TUE, in control and saline conditions at the four proposed intervals: 2–9, 2–6, 6–9 and 9–13 days after treatment. Because the derived indices already take the treatment into account, they are not applied to the above-suggested model. For the interaction model, the significant $P$ values for marker–trait associations were determined using the Bonferroni-adjusted threshold of $\alpha = 0.05$, which corresponded to $P = 8.99 \times 10^{-6}$, $2.57 \times 10^{-6}$ and $3.02 \times 10^{-6}$ for the *INDAUS*, *indica* and *aus* subpopulations, respectively.

Because the use of the conventional MLM with TASSEL is less statistically powerful than our interaction model, we found that neither the Bonferroni nor false discovery rate[53,54] nor the Benjamini–Hochberg showed significant $P$ values. Although other previous rice GWAS studies have used a significant threshold of $P = 10^{-5}$ for the conventional MLM using TASSEL[24,25]. We have retained this as only a suggestive threshold and present these outputs in Supplementary Fig. 5g–i and Supplementary Table 4.

For both models, the minus $\log_{10}$ of the genome-wide observed $P$ values were displayed in Manhattan plots (Supplementary Fig. 5), using the *qqman* package for R[55].

The LD statistic $r^2$ was based on genotype allele counts and was estimated for pairs of SNP loci using Plink software (http://pngu.mgh.harvard.edu/purcell/plink/)[56]. Local LD surrounding the significant SNPs for all the traits is presented in Supplementary Data 1.

No computational approaches were used to reduce computational time because the use of splines and further interval selection significantly reduced the number of calculation time points for our dynamic analyses. For TASSEL, the data were processed in parallel in a cluster environment, while for the interaction model in *ASReml-R*, the data were processed using a laptop computer.

**Data availability.** The data from which the traits were calculated, the trait values, the codes used in producing the trait values and the code for the interaction model used for GWAS analyses are all available in Dryad (http://datadryad.org/, doi:10.5061/dryad.3118j). The authors declare that all other data supporting the findings of this study are available within the article and its Supplementary Information files (or are available from the corresponding author on request).

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

## Acknowledgements

The research reported in this publication was supported by funding from King Abdullah University of Science and Technology. We thank Michael Dingkuhn, Brigitte Courtois, Kenneth McNally and Julie Mae Pasuquin from the Global Rice Phenotyping Network. All seed material was kindly provided by the International Rice Genebank (International Rice Research Institute, Philippines). We thank Professor Susan McCouch (Cornell University) for providing the 'HDRA 700kK SNPs' data for the GWAS and analytical comments. We thank all members at The Plant Accelerator: Dr Rachel Burton, Helli Meinecke, Dr Trevor Garnett, Dr Alex Garcia, Richard Norrish, Dr Guntur Tanjung, George Sainsbury, Evi Guidolin, Robin Hosking, Lidia Mischis, Nicky Bond, Sepideh Azizi Taramsary, Kate Dowling and Fiona Groskreutz for providing technical support in the collection of phenotypic data. The Plant Accelerator, Australian Plant Phenomics Facility, is supported under the National Collaborative Research Infrastructure Strategy

of the Australian Government. We thank Heno Hwang for scientific illustrations of the Smarthouses illustration of pot design. We also thank Bo Li and Inês Silva Pires for critical comments.

## Author contributions

N.A.-T. performed most of the data analyses and wrote the manuscript. C.B. designed the experiments, performed the spatial correction, and conceived of and developed the statistical analyses for the phenotypic data. H.O. developed the interaction model for association analyses. B.B. and N.A.-T. phenotyped the plants and analysed the imaging data. N.A.-T., Y.S.H., H.O. and S.N. performed the genotypic analyses and the analyses of GWAS. S.S. and S.M.S. assisted with the phenotypic data analyses. M.T. and S.N. contributed to the original concept of the project and supervised the study. S.N. conceived the project and its components. All authors have read and contributed to the manuscript.

## Additional information

**Competing financial interests:** The authors declare no competing financial interests.

