## [Peer Review File · Nature Communications]

Reviewers' comments:

Reviewer #1 (Remarks to the Author):

In this manuscript, a GWAS for salinity tolerance in a panel consisting of indica and aus rice genotypes is conducted. There are two novel aspects of this work: i.) smoothing splines are used to model the phenotypes across 13 different time points, and ii.) the GWAS model includes additional fixed effects for treatment and the interaction between treatment and markers. Some novel loci for transpiration use efficiency are detected using this approach.

Overall, I think that this work is solid, especially with respect to the statistical analysis that was conducted. Additionally, I think that the manuscript is generally well written (although it would not hurt to go through another round of revisions). My main constructive criticisms of the manuscript are as follows:

- 1.) I disagree the thresholds that were used to determine statistical significance. I strongly suggest using the Benjamini-Hochberg procedure that controls the false discovery rate at 5%.
- 2.) I felt that there were too many acronyms in the manuscript, some of which (for example, DAST; I would just say "days after salting") are unnecessary. I think that the manuscript would be easier to follow if the number of acronyms were reduced.
- 3.) Because some novel mathematical and statistical approaches are being tried here, I think it would be beneficial to have the Materials and Methods before the Results. If this is not possible, then I suggest adding some more details about the novel statistical analyses in the Results.
- 4.) I think that local linkage disequilibrium (LD) surrounding peak SNPs identified in the GWAS needs to be further explored and described in the results. Although there are plausible candidate genes in the vicinity of SNPs with peak associations, it is important to explore the local LD to see if SNPs surrounding these candidate genes are in LD with the peak SNPs.

Here are some notes I took while reading through the manuscript:

Line 107: "an unbiased analysis". Please explain in the text what is meant by "unbiased".

Lines 112-113: I think it is unnecessary to include the fitted regression model. I think that reporting Pearson correlation coefficients is sufficient.

Line 163: What does "third order derived trait" mean?

Line 178: "GWAS" - Please make sure that all acronyms are spelled out the first time they appear in the manuscript.

Lines 188-189 - Please cite Yu et al. (2006) when describing the unified mixed linear model:
Yu, J. M., G. Pressoir, W. H. Briggs, I. V. Bi, M. Yamasaki et al.,
2006 A unified mixed-model method for association mapping
that accounts for multiple levels of relatedness. Nat. Genet. 38:
203-208.

Line 201: "qRGR(STI)3-1" I suggest giving the SNP name of all peak SNPs. In my opinion, reporting the names of peak SNPs might be more useful for other researchers who want to use these results for further study.

Line 213: What is TUE-TOL and TUE-W3?

Line 223: For " $p < 0.0001$ ". Please use α (i.e., the type I error rate) instead of a P-value.

For all figure legends: Please spell out all acronyms. Basically, all figure legends (and table captions) should "stand on their own", in the sense that the reader does not have to go back to the manuscript text to figure out what the acronyms mean.

Lines 476 - 479: In my opinion, these equations are presented in a sloppy fashion. Please use proper statistical notation when presenting statistical models. For example, the new mixed linear model presented in between lines 518 and 519 can be used as a template for rewriting the equations on lines 476-479.

Lines 522-531: Was anything like P3D, EMMAX, or GEMMA used? If not, were any computational approaches used to speed up computational time? Regardless, how long did it take to run a GWAS using this model on one trait?

Lines 533-534: To reiterate, the Benjamini-Hochberg procedure should be used to control the false discovery rate. The thresholds that control either the type I error rate or false discovery rate should be used instead of somewhat arbitrary "suggestive" thresholds.

Reviewer #2 (Remarks to the Author):

The paper on "New salinity tolerance loci revealed in rice using high-throughput non-invasive

phenotyping" is nicely written paper with far-reaching conclusion that are not well supported using concrete evidences. Authors have used automated HTP non-evasive system for the TR, TUE and RGR but the output of this study does not make it very novel in terms of significance and impact on advancement for the development of salinity tolerant rice genotypes with high yield. My comments on the paper are :

1. Authors claimed that "This study provides new insights intoand will contribute to crop improvement". The study involved 24 days old seedlings when they start gaining tolerance and the degree of discrimination between tolerant and sensitive becomes less comparing when 7-10 days old seedling is used for screening.
2. Another point which is also related with extrapolated claim for crop (rice) improvement, rice is extremely sensitive at early seedling stage and reproductive stage (especially about 10 days + and - to booting), but not very sensitive at other stages. Tolerance at seedling stage does not translate into grain yield (ultimate for rice improvement) but tolerance to reproductive stage definitely translates into grain yield. There are numerous studies that explain conclusively that there is very poor association of seedling stage salinity tolerance vs reproductive stage salinity tolerance or with grain yield. There are different set of genes / QTLs that govern seedling and reproductive stage salinity tolerance. What I want to emphasize that there is not much importance of this study to the scientists in this field.
3. Authors tried to over-emphasize the importance of osmotic component of salt stress in rice with non-conclusive evidences or without concrete evidences at many places in MS (line 118-121, 320-323). Indeed osmotic stress depends much more upon species the tolerance level of genotypes. In rice, the osmotic phase is very short lived and overtakes by ionic toxicity with few hours. There are plenty of studies with hourly and daily salt uptake in rice that show the short-lived osmotic component of stress. But in this study, authors did not use few buffer plants to show that their leaves did not have the salt concentration until 2-6 days. So suggestive evidences are not good-enough for claiming the contrary findings.

I don't feel that this paper will influence thinking or make any big impact in the relevant field and has merit to be published in Nature communications.

Reviewer #3 (Remarks to the Author):

The present study utilized high-throughput phenotyping technology and GWAS to analyze rice plant salinity tolerance. The paper provides a straight-forward to follow and at the same time very detailed description of the utilized methods. Of high interest to the research community is likely the consideration of complex to analyze traits such as relative growth rates, transpiration rates and transpiration use efficiency throughout the plant development under different treatments. The presented data and methodology seems to be valid, the cleansing of the data is

explained in sufficient detail and the overall quality of the presentation is high. Utilized statistical methods are referenced and standard deviations, confidence intervals etc. are tabulated as needed. The 'new salinity tolerance loci' seem to be complex and only significant at certain time periods. Detailed follow-up experiments or replications of the experiment are eased by the in-depth explanations, e.g. on exact pot design, soil covering using gravel and plastic mats and plant placement regime. To improve the paper it is suggested to give more details on how and why six derived indices (lines 190 and following, and 466 and following) have been chosen out of the 24 in suppl. table 3). It is not detailed, how the plant shoot area (PSA) is exactly considered. Is it the sum of the projected areas from side and top, or is it a volume estimation? As this data and information is a essential for the analysis, it should be shortly explained in the paper and not only available from reference literature. In line 173 a 'box plot' in figure 2e is mentioned, but figure 2e is a table (should be included as a table). Figure 2c and d should use the same scaling for the y axis, to ease the comparison of the two genotypes. In the paper a 'new' association model (lines 218 and 517) is explained. It was not clear, if this model has been created newly for the paper or if it has been just recently used in related studies, why and what is new? In lines 297 to 303, the advantages of the spline-model are explained in detail, but no disadvantage was mentioned. The advantage of no a priori assumption on the shape of the curve may in some situation also be a disadvantage, as it makes model-based predictions on the data more difficult. The advantage of having just a few clearly defined curve determinants describing the overall growth dynamic e.g. in the logistic growth model are not mentioned. On the experimental procedures, it was not clear if there was a larger biomass variability, and if this affects the weighing data and thus the watering regime (lines 388 and following). As transpiration is largely affected by illumination intensity, it is suggested to think about considering in the future sensor-based illumination and temperature data for the analysis. The provided link to the image data (line 430) did not work in my tests. Before publication this problem should be fixed. Overall the paper is clearly written and considers a wide range of appropriate reference literature. After minor revision it is recommended for publication.

**Reviewers' comments on manuscript entitled "New salinity tolerance loci revealed in rice using high-**
**throughput non-invasive phenotyping"- NCOMMS-16-03546-T**

**We thank the referees for their comments to which we have replied to below.**

**Reviewer #1 (Remarks to the Author):**

*In this manuscript, a GWAS for salinity tolerance in a panel consisting of indica and aus rice genotypes is*
*conducted. There are two novel aspects of this work: i.) smoothing splines are used to model the*
*phenotypes across 13 different time points, and ii.) the GWAS model includes additional fixed effects for*
*treatment and the interaction between treatment and markers. Some novel loci for transpiration use*
*efficiency are detected using this approach.*

*Overall, I think that this work is solid, especially with respect to the statistical analysis that was*
*conducted. Additionally, I think that the manuscript is generally well written (although it would not hurt*
*to go through another round of revisions). My main constructive criticisms of the manuscript are as*
*follows:*

*1.) I disagree the thresholds that were used to determine statistical significance. I strongly suggest using*
*the Benjamini-Hochberg procedure that controls the false discovery rate at 5%.*

*The suggestive threshold of $p = 10^{-5}$ as a cut-off for significance has recently been used in other rice*
*studies such as Crowel et al. (2016: *Nat. Comm.* 7:10527) and Rebolledo et al. (2015: *J. Exp. Bot.* 66:*
*5555-5566). The Crowel et al. (2016) study is analogous to ours because the same genotypic data (700k*
*SNP HDRA) is used, and because a conventional mixed linear model (MLM) is also used; while the*
*Rebolledo et al. (2015) study used a lower threshold of 1×10^{-4} , and also performed the association*
*analysis using the conventional MLM approach with TASSEL software, just as we did. Therefore, we*
*believe that the suggestive significant threshold of $p = 10^{-5}$ is an appropriate level for the conventional*
*MLM we used in this study. Nevertheless, we followed the referee's suggestion and calculated the*
*Benjamini-Hochberg critical value for each p-value to control the false discovery rate at the 5% level for*
*the conventional MLM model using TASSEL. The Benjamini-Hochberg cut-off value to determine*
*significance of a SNP was taken as the largest p-value, where the p-value was less than the Benjamini-*
*Hochberg critical value. In addition, we investigated the significance of a SNP using the False Discovery*
*Rate (FDR) and Bonferroni correction. We found that neither the Benjamini-Hochberg, nor the FDR, nor*
*the Bonferroni thresholds showed significant p-values for the conventional MLM. For our (more*
*powerful) proposed interaction model, which was the main focus of the paper, we used the Bonferroni*
*threshold at 5% significance to determine SNP significance and these results are the ones presented in*
*the manuscript. To be consistent with the significance testing of both the conventional and interaction*
*models, we have now included the Bonferroni results in the text for both models and clarified the*
*manuscript text about the difference in results found using the two models specifically by showing that*
*the interaction model provided a higher statistical power.*

2.) I felt that there were too many acronyms in the manuscript, some of which (for example, DAST; I
would just say "days after salting") are unnecessary. I think that the manuscript would be easier to
follow if the number of acronyms were reduced.

> We agree with this comment and we reduced the number of acronyms. We hope that this makes the
text clearer.

3.) Because some novel mathematical and statistical approaches are being tried here, I think it would be
beneficial to have the Materials and Methods before the Results. If this is not possible, then I suggest
adding some more details about the novel statistical analyses in the Results.

> These are the rules of *Nature Communications*. However, we tried to address the referee's comment
by extending the details in the Results section.

4.) I think that local linkage disequilibrium (LD) surrounding peak SNPs identified in the GWAS needs to
be further explored and described in the results. Although there are plausible candidate genes in the
vicinity of SNPs with peak associations, it is important to explore the local LD to see if SNPs surrounding
these candidate genes are in LD with the peak SNPs.

> We have performed these analyses before and we present the results for all traits and intervals in the
supplementary materials (Supplementary Data 1). To address the referee's comment, we have added to
the manuscript the local LD results for transpiration use efficiency (TUE) in a new table (Table 2), under
the section "Candidate genes underlying QTLs suggest signaling is involved in early response to salt
stress".

Here are some notes I took while reading through the manuscript:

Line 107: "an unbiased analysis". Please explain in the text what is meant by "unbiased".

> The use of the word "unbiased" is further explained in the discussion (lines 306-309) as we realized
that this remark needs elaboration, because it is more of a discussion point than a result.

Lines 112-113: I think it is unnecessary to include the fitted regression model. I think that reporting
Pearson correlation coefficients is sufficient.

> We have used the Pearson's correlation and we have presented these results in Supplementary Fig. 4.
In the revised manuscript we only report the r^2 value.

Line 163: What does "third order derived trait" mean?

> We have explained why transpiration use efficiency (TUE) is a third order derived trait on lines 161-
163, namely because TUE is estimated from transpiration and growth from projected shoot area (PSA),
and these traits are also estimated from water loss and pixel count, respectively.

Line 178: "GWAS" - Please make sure that all acronyms are spelled out the first time they appear in the
manuscript.

> We have introduced the acronym GWAS in the last paragraph of the introduction, specifically on line
70 of the revised manuscript. We have been careful to ensure all acronyms are spelt out in full at the
time of their first use.

*Lines 188-189 - Please cite Yu et al. (2006) when describing the unified mixed linear model:*
*Yu, J. M., G. Pressoir, W. H. Briggs, I. V. Bi, M. Yamasaki et al.,2006 A unified mixed-model method for*
*association mapping that accounts for multiple levels of relatedness. Nat. Genet. 38: 203-208.*
*> Thank you, this reference was indeed missing. We have introduced this reference in line 187 and also*
*in the methods section in line 565.*
*Line 201: "qRGR(STI)3-1" I suggest giving the SNP name of all peak SNPs. In my opinion, reporting the*
*names of peak SNPs might be more useful for other researchers who want to use these results for further*
*study.*
*> We have changed the name of all the SNPs included in the peaks to their SNP ID or we mention the LD*
*region surrounding the most significant SNP.*
*Line 213: What is TUE-TOL and TUE-W3?*
*>These indices are based on the stress tolerance (TOL) index coined by Fernandez (1992) and proposed*
*earlier by Rosielle and Hamblin (1981), as reported in Table 1 of the main manuscript. We have now*
*removed W3 from the manuscript.*
*Line 223: For "p<0.0001". Please use α (i.e., the type I error rate) instead of a P-value.*
*> We have amended this in the text.*
*For all figure legends: Please spell out all acronyms. Basically, all figure legends (and table captions)*
*should "stand on their own", in the sense that the reader does not have to go back to the manuscript text*
*to figure out what the acronyms mean.*
*> We have gone through all the figure legends and corrected them accordingly.*
*Lines 476 - 479: In my opinion, these equations are presented in a sloppy fashion. Please use proper*
*statistical notation when presenting statistical models. For example, the new mixed linear model*
*presented in between lines 518 and 519 can be used as a template for rewriting the equations on lines*
*476-479.*
*> We have rewritten part of the methods section "Spatial correction of phenotypic analysis" and the*
*format of all equations is now consistent.*
*Lines 522-531: Was anything like P3D, EMMAX, or GEMMA used? If not, were any computational*
*approaches used to speed up computational time? Regardless, how long did it take to run a GWAS using*
*this model on one trait?*
*> No, we did not use these computational approaches to hasten computational time. For TASSEL, the*
*data was processed in parallel in a cluster environment, while for the interaction model in ASReml-R the*
*data were processed using a laptop. We have improved the text in the methods section to address the*
*referee's comment to further explain the computational time spent undertaking the analyses (line 599-*
*603).*

*Lines 533-534: To reiterate, the Benjamini-Hochberg procedure should be used to control the false*
*discovery rate. The thresholds that control either the type I error rate or false discovery rate should be*
*used instead of somewhat arbitrary "suggestive" thresholds.*

> We have replied to this comment in the first point raised by the referee.

**Reviewer #2 (Remarks to the Author):**

*The paper on "New salinity tolerance loci revealed in rice using high-throughput non-invasive*
*phenotyping" is nicely written paper with far-reaching conclusion that are not well supported using*
*concrete evidences. Authors have used automated HTP non-evasive system for the TR, TUE and RGR but*
*the output of this study does not make it very novel in terms of significance and impact on advancement*
*for the development of salinity tolerant rice genotypes with high yield. My comments on the paper are:*

*1. Authors claimed that "This study provides new insights intoand will contribute to crop*
*improvement". The study involved 24 days old seedlings when they start gaining tolerance and the*
*degree of discrimination between tolerant and sensitive becomes less comparing when 7-10 days old*
*seedling is used for screening.*

> To increase the ability of a crop plant to maintain yield under high salinity, it needs to be able to
maintain functions at all stages of its lifecycle. That we studied only one growth phase in no way
invalidates this study, and does not reduce its potential to make significant contributions to increasing
salinity tolerance. We imposed salinity on 29 day-old plants that were at approximately at the 5-leaf
stage, and we have clarified this in the methods section (line 417). Salt-stress was imposed at this stage
because we specifically wanted to quantify transpiration rate and transpiration use efficiency. To obtain
rates of transpiration that were significantly greater than the evaporation of water from the soil surface,
we needed to impose salinity on bigger plants (older plants) to collect individual plant water loss data. It
is known that yield is also affected when plants are exposed to salinity at the vegetative stage, affecting
parameters such as tiller number per plant and duration of stress (Zeng et al. (2001), Agricultural Water
Management 48:191-206; Zeng and Shannon (2000), Crop Science 40: 996-1003; Zeng et al. (2002),
Euphytica 127: 235-245). We acknowledge that the main vegetative stage is less sensitive to salinity
compared to the early seedling stage (with 2 to 3 leaves); however, our data clearly shows a significant
effect of salinity on plant growth and transpiration and significant variation in responses to salinity (as
presented in Figures 1 and 2). We have introduced a new paragraph at the end of the discussion to
address the referee's concern (line 341-352). We have clarified the importance of studying rice salinity
tolerance at the vegetative stage to further quantify transpiration use efficiency to assist crop
improvement.

*2. Another point which is also related with extrapolated claim for crop (rice) improvement, rice is*
*extremely sensitive at early seedling stage and reproductive stage (especially about 10 days + and - to*
*booting), but not very sensitive at other stages. Tolerance at seedling stage does not translate into grain*
*yield (ultimate for rice improvement) but tolerance to reproductive stage definitely translates into grain*
*yield. There are numerous studies that explain conclusively that there is very poor association of seedling*

*stage salinity tolerance vs reproductive stage salinity tolerance or with grain yield. There are different set*
*of genes / QTLs that govern seedling and reproductive stage salinity tolerance. What I want to*
*emphasize that there is not much importance of this study to the scientists in this field.*

> The referee is correct about the poor correlation between salinity tolerance in the seedling and
reproductive stages of growth. Nevertheless, as we explained in the previous paragraph, we have
imposed salinity from when plants were 29 days old, and the stress was imposed for 13 days (the
experiment ended when plants were 42 days old), which translates to the salt being imposed during
vegetative growth, specifically during the tillering stage (as seen in ricepedia [http://ricepedia.org/rice-](http://ricepedia.org/rice-as-a-plant/growth-phases)
[as-a-plant/growth-phases](http://ricepedia.org/rice-as-a-plant/growth-phases)). Rice yield is a result of several complex traits, such as sowing density,
number of panicles per unit land area, number of grain produced per panicle (i.e. panicle length,
number of florets per panicle, fertility rate) and average weight of individual grains (i.e. panicle weight,
total seed weight and 1000 seed weight). One important yield component, potential number of panicles
178 per plant (i.e. tiller number per plant), is determined by the plant at the time we conducted our
experiment. Moreover, IR64, a typical *indica* accession present in our *indica* panel, has the vegetative
stage duration of approximately 45 days
([http://www.knowledgebank.irri.org/ericeproduction/0.2. Growth stages of the rice plant.htm](http://www.knowledgebank.irri.org/ericeproduction/0.2.Growth%20stages%20of%20the%20rice%20plant.htm)),
which translates into some of the accessions, such as IR64, being stressed during the initiation of the
reproductive stage. We are confident that the results of our salinity experiment are relevant to
processes that significantly affect rice yield, and that the novel loci we have discovered, when
incorporated into breeding programs, will impact on salinity tolerance of rice. Furthermore, our work
provides novel scientific insights into fundamental processes in plants and the effects of salinity on
these.

*3. Authors tried to over-emphasize the importance of osmotic component of salt stress in rice with non-*
*conclusive evidences or without concrete evidences at many places in MS (line 118-121, 320-323). Indeed*
*osmotic stress depends much more upon species the tolerance level of genotypes. In rice, the osmotic*
*phase is very short lived and overtakes by ionic toxicity with few hours. There are plenty of studies with*
*hourly and daily salt uptake in rice that show the short-lived osmotic component of stress. But in this*
*study, authors did not use few buffer plants to show that their leaves did not have the salt concentration*
*until 2-6 days. So suggestive evidences are not good-enough for claiming the contrary findings.*

> We disagree with the referee's assertion that the osmotic phase is very short-lived and is overtaken by
the ionic phase within a few hours. To our knowledge, we do not know studies where these kinds of
measurement, especially in rice, have been undertaken. In fact, there is no evidence that the so-called
osmotic phase is "overtaken" at all. We suspect this will depend on many factors, such as the
concentration of salts in the external solution, transpiration rates, and the effects of other stresses on
the plant. The ionic effects do not require simply the accumulation of ions - the ions have to accumulate
to a sufficient extent and for a sufficiently long time for the ionic effects to become evident. Given our
study is focusing on the effects of salinity on the production of new leaves, and given the ionic phase is
often defined by the effects of salt accumulation on acceleration of the senescence of the older leaves,
the referee's argument could rapidly become a semantic one. We could, if the editors want, formulate a
new term to ensure that we define that the processes being studied in the current manuscript are
clearly focusing on the growth of new leaves and not on the death of old leaves. However, we think this

semantic is unlikely to be helpful, and, to us, at least, it is self-evident that we are focusing on a process
related to early salinity response, which is clearly distinct from the one related to the longer-term
premature senescence of older leaves (the process generally accepted in the literature as being the one
primarily related to the so-called “ionic phase”). We have also added four lines of text addressing this
issue in the Discussion, lines 340-343.

*I don't feel that this paper will influence thinking or make any big impact in the relevant field and has
merit to be published in Nature communications.*

> We think that this referee has misunderstood some of the primary points of this work and disregarded
the novelty of the methods presented in the manuscript, as evidenced by all three of his/her comments.
We hope that our explanations above are sufficiently clear to enable a discounting of the final opinion
proffered by this referee.

**Reviewer #3 (Remarks to the Author):**

*The present study utilized high-throughput phenotyping technology and GWAS to analyze rice plant
salinity tolerance. The paper provides a straight-forward to follow and at the same time very detailed
description of the utilized methods. Of high interest to the research community is likely the consideration
of complex to analyze traits such as relative growth rates, transpiration rates and transpiration use
efficiency throughout the plant development under different treatments. The presented data and
methodology seems to be valid, the cleansing of the data is explained in sufficient detail and the overall
quality of the presentation is high. Utilized statistical methods are referenced and standard deviations,
confidence intervals etc. are tabulated as needed. The 'new salinity tolerance loci' seem to be complex
and only significant at certain time periods. Detailed follow-up experiments or replications of the
experiment are eased by the in-depth explanations, e.g. on exact pot design, soil covering using gravel
and plastic mats and plant placement regime.*

*To improve the paper it is suggested to give more details on how and why six derived indices (lines 190
and following, and 466 and following) have been chosen out of the 24 in suppl. table 3).*

> We have improved the text to further explain the number of indices, which are now five (line 187-
191). Moreover, we have removed W3 from our table results as previously mentioned, which has
reduced the number of indices used to five.

*It is not detailed, how the plant shoot area (PSA) is exactly considered. Is it the sum of the projected
areas from side and top, or is it a volume estimation? As this data and information is a essential for the
analysis, it should be shortly explained in the paper and not only available from reference literature.*

> We have improved the methods section to address the referee's comment (line 456-459 and 472-475).
The projected shoot area (PSA) is estimated based on the sum of all three images (two side views and
the top view). There is quite a large volume of literature now published on this, from previous work
undertaken in The Plant Accelerator.

*In line 173 a 'box plot' in figure 2e is mentioned, but figure 2e is a table (should be included as a table).*

> This was a mistake and we have corrected it.

*Figure 2c and d should use the same scaling for the y axis, to ease the comparison of the two genotypes.*

> The scale is the same; maybe the referee had difficulties on visualizing this figure?

*In the paper a 'new' association model (lines 218 and 517) is explained. It was not clear, if this model has*
*been created newly for the paper or if it has been just recently used in related studies, why and what is*
*new?*

> This model was newly created for this paper. We have not seen any similar association model being
used to address stress (biotic or abiotic) treatment. The approach considers all the data (salt and
control) as a single response, with treatment (salt vs. control) and its interaction with markers as
cofactors in the model. This is opposed to the standard approach of GWAS available via packages such
as TASSEL, where the salt and treatment data need to be considered separately as different responses.
Thus, in this new approach, in addition to the usual marker cofactor of a standard GWAS, a term for
treatment and the interaction between marker and treatment are also included. This means that we can
not only determine markers that are associated with the response, but we can look for markers that are
salt specific through parameterization of the model by considering the interaction term.

*In lines 297 to 303, the advantages of the spline-model are explained in detail, but no disadvantage was*
*mentioned. The advantage of no a priori assumption on the shape of the curve may in some situation*
*also be a disadvantage, as it makes model-based predictions on the data more difficult. The advantage*
*of having just a few clearly defined curve determinants describing the overall growth dynamic e.g. in the*
*logistic growth model are not mentioned.*

> We have improved the discussion to address the referee's comment, and we further describe the
advantages of logistic/exponential models and the advantages of splines (line 296-328).

*On the experimental procedures, it was not clear if there was a larger biomass variability, and if this*
*affects the weighing data and thus the watering regime (lines 388 and following).*

> We have addressed the referee's comment by improving the methods section (line 416-419) and
explaining that the small absolute difference in the overall shoot biomass, as well as between the
treatments (when compared to the overall pot weight, including the water), makes the larger biomass
variability negligible; thus, the water levels used were the same in both treatments.

*As transpiration is largely affected by illumination intensity, it is suggested to think about considering in*
*the future sensor-based illumination and temperature data for the analysis.*

> We acknowledge this comment. The Plant Accelerator now has sensors installed between every lane
to measure the humidity, temperature and light. However, at the time of this experiment, the sensors
were still not operational. Future studies at The Plant Accelerator will greatly benefit from these data
records.

*The provided link to the image data (line 430) did not work in my tests. Before publication this problem*
*should be fixed.*

> The link was going to become operational once the DOI number is attributed. However, we have now
made the links to the image data live and you can access the complete RGB data set for indica and aus
panels online through The Plant Accelerator® data portal.

*Overall the paper is clearly written and considers a wide range of appropriate reference literature. After*
*minor revision it is recommended for publication.*

Reviewers' Comments:

Reviewer #1 (Remarks to the Author)

In my opinion, the authors did a great job of addressing the majority of my (and the other reviewer's) comments in this manuscript. I have only two further suggestions (one of them is extremely minor) about the statistical aspects of the paper:

1.) For the "suggestive" thresholds (of $P = 1.0 \times 10^{-5}$) used to designate "significant" GWAS results (particularly in the section between lines 176-248). I completely disagree with the argument of "this threshold has been used in previous studies so therefore it is ok to use it for our study". In general, just because an approach is used in the literature does not mean that it is correct. Moreover, equating this suggestive threshold to statistical significance (as done in Line 217) is both misleading to the reader and statistically incorrect. Please consistently use a significance threshold that controls either a type I error rate (e.g., Bonferroni) or a false discovery rate (e.g., the Benjamini-Hochberg procedure) for both the traditional MLM and the new statistical model incorporating marker by treatment interactions; at the very least, an "apples to apples" comparison can then be made between the two models. Also, please indicate in the captions to Table 2 and Figure 3 what criterion (e.g., Bonferroni) was used to determine statistical significance.

From a quantitative genetics perspective, I think that a lot of impressive work was done in this manuscript. Please do not diminish this study by using a significance threshold that is not based on well-grounded statistical theory.

2.) Lines 111-112: r^2 is actually the squared Pearson correlation coefficient. Please make this clear in the text.

Reviewer #2 (Remarks to the Author)

This is the revised MS submitted by the authors and i am satisfied by the response for the first two points but i still standby with my third comment on over-emphasis on osmotic component of salt stress. There is no concrete evidence that it last for a long. I am attaching one classical work by Yeo et al. 1991 (JXB) that clearly showed the short lived osmotic response in rice overcome by ionic component of stress using the real-time experiment on leaf elongation and ionic uptake. The displacement transducer very nicely showed the time-course of leaf elongation under culture solution with NaCl, KCl and mannitol.

Reviewer #3 (Remarks to the Author)

The issues mentioned in my previous review have been addressed to my satisfaction.

One open point remains, the scaling of the y axis in image 1 c and d should be the same, to ease the comparisons of the two genotypes. In my previous comments incorrectly image 2 was mentioned.

After this minor correction from my point of view the paper is acceptable for publication.

Reviewers' comments on manuscript entitled "Salinity tolerance loci revealed in rice using high-throughput non-invasive phenotyping"-NCOMMS-16-03546-T

We thank the referees for their comments to which we have replied to below.

Reviewer #1 (Remarks to the Author):

In my opinion, the authors did a great job of addressing the majority of my (and the other reviewer's) comments in this manuscript. I have only two further suggestions (one of them is extremely minor) about the statistical aspects of the paper:

1.) For the "suggestive" thresholds (of $P = 1.0 \times 10^{-5}$) used to designate "significant" GWAS results (particularly in the section between lines 176-248). I completely disagree with the argument of "this threshold has been used in previous studies so therefore it is ok to use it for our study". In general, just because an approach is used in the literature does not mean that it is correct. Moreover, equating this suggestive threshold to statistical significance (as done in Line 217) is both misleading to the reader and statistically incorrect. Please consistently use a significance threshold that controls either a type I error rate (e.g., Bonferroni) or a false discovery rate (e.g., the Benjamini-Hochberg procedure) for both the traditional MLM and the new statistical model incorporating marker by treatment interactions; at the very least, an "apples to apples" comparison can then be made between the two models.

>We agree with the referee's comments, and followed the editors suggestion to move the conventional MLM results to the supplementary information file and clearly stated that no significant p-values were found with conventional MLM after Bonferroni/BH correction. We changed the results and discussion to fit accordingly. We removed the discussion for the conventional MLM analysis in lines 191-215 (and 279-281) and focused entirely on the associations found with the interaction model results.

Also, please indicate in the captions to Table 2 and Figure 3 what criterion (e.g., Bonferroni) was used to determine statistical significance.

> We have now indicated the criterion used for Table 2 and Figure 3 in the main text. Statistical significance of the GWAS associations were determined using the Bonferroni-adjusted threshold of $\alpha = 0.05$, which corresponded to $P = 8.99 \times 10^{-6}$, 2.57×10^{-6} and 3.02×10^{-6} for the *INDAUS*, *indica* and *aus* sub-populations respectively.

From a quantitative genetics perspective, I think that a lot of impressive work was done in this manuscript. Please do not diminish this study by using a significance threshold that is not based on well-grounded statistical theory.

We agree with the referee and thank them for their comments.

2.) Lines 111-112: r^2 is actually the squared Pearson correlation coefficient. Please make this clear in the text.

> We have used the Pearson correlation and have now made this clear in the main text.

Reviewer #2 (Remarks to the Author):

This is the revised MS submitted by the authors and i am satisfied by the response for the first two points but i still standby with my third comment on over-emphasis on osmotic component of salt stress. There is no concrete evidence that it last for a long. I am attaching one classical work by Yeo et al. 1991 (JXB) that clearly showed the short lived osmotic response in rice overcome by ionic component of stress using the real-time experiment on leaf elongation and ionic uptake. The displacement transducer very nicely showed the time-course of leaf elongation under culture solution with NaCl, KCl and mannitol.

>It is clear that the experiments undertaken by Yeo et al (1991) and us are very different. The Yeo et al (1991) paper describes experiments in which the effects of salinity on young rice plants was studied. In our work, the plants were much older (salt was added 29 days after germination, not 15 days), were grown hypoxically in soil (not in aerated hydroponics), and the salt was added more gently (being infiltrated slowly into the soil, not added directly to the hydroponics solution which is in direct contact with the roots).

The differences between the experiments is most starkly indicated by the fact that the Yeo et al paper is concentrating on the death of leaves of relatively young, small plants - whereas in our work, no leaf death was seen at all. The difference in effect is most likely to be primarily due to the differences in the ages of the plants, and possibly also in the differences in substrate in which the plants were grown. In any case, given the primary symptoms of ionic toxicity is premature death of older leaves, and given we did not see any death of older leaves (let alone premature death), we are confident that the primary effects we were seeing were not related to ionic toxicity. The clear inhibition of production of new leaves that we quantify very clearly is consistent with the primary effects of salinity in our study being due mostly to so-called osmotic effects.

In addition, one of the primary genetic bases for variation in ionic toxicity is now well documented in rice, being at a locus on chromosome 1, associated with *OsHKT1;5*. In our study, none of the QTLs we observed overlapped with this gene, nor any of the other 8 members of this gene family found in most rice genomes. This also provides evidence that is consistent with the suggestion that the effects we are studying are due to processes independent of ion accumulation.

To present this last argument in the text, we have added the following sentence to the end of the paragraph in question, on page 13, old line 343:
“That the early phase is likely to be independent of shoot ion accumulation (and so is the “osmotic phase”) is also supported by the absence of QTLs identified in previous studies (Ren et al., 2005) that affect leaf ion accumulation, in particular the locus containing *OsHKT1;5*, on chromosome 1.”

Reviewer #3 (Remarks to the Author):

The issues mentioned in my previous review have been addressed to my satisfaction.

One open point remains, the scaling of the y axis in image 1 c and d should be the same, to ease the comparisons of the two genotypes. In my previous comments incorrectly image 2 was mentioned.

> We amended the images and they now have the same scaling of the y axis.

After this minor correction from my point of view the paper is acceptable for publication.